# Underwater Spiral Wave Sound Source Based on Phased Array with Three Transducers

**DOI:** 10.3390/s19143192

**Published:** 2019-07-19

**Authors:** Wei Lu, Rongzhen Guo, Yu Lan, Hao Sun, Shichang Li, Tianfang Zhou

**Affiliations:** 1Acoustic Science and Technology Laboratory, Harbin Engineering University, Harbin 150001, China; 2Key Laboratory of Marine Information Acquisition and Security (Harbin Engineering University), Ministry of Industry and Information Technology, Harbin 150001, China; 3College of Underwater Acoustic Engineering, Harbin Engineering University, Harbin 150001, China

**Keywords:** spiral sound wave, phased array, phase directivity, finite element

## Abstract

This paper realizes an underwater spiral wave sound source by using three omni-directional spherical transducers with three different phases. The pressure distribution of the sound field for a phased array is derived using the superposition theory of sound field. The generation of spiral wave field is presented, the relationship between the performance of phased array sound field and the array parameters is analyzed, and also verified by the finite element method (FEM). A spiral wave sound source with three spherical piezoelectric ceramic transducers is then designed and fabricated based on FEM simulation, and the performance of the sound source is analyzed. Measurements are made in a reverberation pool, and the result shows that the fabricated spiral wave sound source is capable of producing a spiral sound wave. Under a frequency of 3.5 kHz, the phase directivity has a fluctuation of ±21°, and the amplitude directivity range is 4.3 dB, which verifies the realization of the spiral wave sound source.

## 1. Introduction

As radio-based navigation is not available in deep seas, underwater acoustic navigation becomes significantly important, especially for autonomous underwater vehicles (AUVs). Acoustic navigation can be performed by long baseline, ultra-short baseline, and recently, a spiral wave method. Among these methods, navigation with spiral wave beacons offers the unique advantage of utilizing limited resources: only one hydrophone is required in an AUV, unlike the long baseline method which requires emitting equipment and the ultra-short baseline method which requires multiple hydrophones to form an array [1,2].

There are two methods to produce a spiral sound wave. One way is to use in-phase radial vibration of a spiral radiation surface. The other way is to use a circular array, where oscillators vibrate with different phases. The first method includes, the “Physical Spiral” transducer proposed by Dzikowicz and Hefner [3,4,5]. This transducer uses 1–3 piezocomposite material formed around a spiral-shaped backing to emit the spiral wave. The second method includes, the “Phase Spiral” transducer, also proposed by Dzikowicz and Hefner [4,5], the Cylindrical Transducer with divided electrodes proposed by Brown [6,7,8], and the Spiral Sound Transducer Based on Longitudinal Vibration Structure proposed by Wei Lu [9]. The “Phased Spiral” transducer also uses 1–3 piezocomposite material and backing, but the electrode is divided into 16 elements. The Cylindrical Transducer with divided electrodes uses a radially polarized piezoceramic hollow cylinder and the electrode is divided into four parts. The Spiral Sound Transducer Based on Longitudinal Vibration Structure consists of eight longitudinal vibrators. By applying different diving voltages to the elements, the three transducers can emit spiral waves. Of the above spiral wave transducers, the Cylindrical Transducer with divided electrodes has the best performance on phase directivity and transmitting voltage response. In addition to large fluctuation in phase directivity, other transducers suffer from other disadvantages. Too many driving voltages needed is one of these disadvantages.

At present, to emit spiral sound waves by an array, it requires a large number of oscillators that vibrate with different phases. The larger the number of oscillators, the more signals are required, the more complex is the driving circuit, and larger the volume and weight of the transducer. This paper, therefore, analyzes a spiral sound source based on phased array with only three omni-directional transducers. Compared with the similar spiral sound sources, the proposed sound source requires the minimum number of transducers. First of all, the sound pressure field produced by a phased array with three transducers is analyzed theoretically. By separating the component of a spiral wave and the component of an interference wave in a sound field, the generation of a spiral wave field is discussed, and the influence of array parameters on the performance of the spiral wave source is analyzed. Then, based on the finite element simulation, a spiral wave sound source based on phased array with three spherical piezoelectric ceramic transducers is designed. The test result indicates that the designed sound source can emit spiral sound waves.

## 2. The Generation of a Spiral Wave Field

### 2.1. Sound Pressure Field Produced by Phased Array with Three Transducers

Figure 1 presents the phased arrays in a polar coordinate system where the three array elements $S_{1}$, $S_{2}$, $S_{3}$ are placed at $\left(d,0\right),\left(d,\frac{2\pi }{3}\right),\left(d,\frac{4\pi }{3}\right)$, respectively.

Assume that the working frequencies and vibration amplitudes of the three array elements are the same, with an initial phase of $0,\frac{2\pi }{3},\frac{4\pi }{3}$ respectively, and all omni-directional. 

When the size of the array element is much smaller than the wavelength, it can be regarded as a point sound source [10,11].

The sound pressure field of the point sound source can be expressed as:(1)$$p_{E}=\frac{A}{r}e^{j\left(\omega t-kr+\varphi_{0}\right)},$$
where, r is the distance from point *P* to the point sound source, ω is the vibration angular frequency of the point sound source, k is the medium wave number, and $\varphi_{0}$ is the initial phase.

According to the superposition theory of sound field [12,13], the sound pressure field produced by the phased array is given as:(2)$$p=\frac{A}{r_{1}}e^{j\left(\omega t-kr_{1}+\frac{0\pi }{3}\right)}+\frac{A}{r_{2}}e^{j\left(\omega t-kr_{2}+\frac{2\pi }{3}\right)}+\frac{A}{r_{3}}e^{j\left(\omega t-kr_{3}+\frac{4\pi }{3}\right)}.$$

There are the following expressions for $r_{1}$, $r_{2}$, $r_{3}$:(3)$$\begin{array}{c} r_{1}=r-d\cos\left(\frac{0}{3}\cdot 2\pi -\varphi \right)\\ r_{2}=r-d\cos\left(\frac{1}{3}\cdot 2\pi -\varphi \right)\\ r_{3}=r-d\cos\left(\frac{2}{3}\cdot 2\pi -\varphi \right) \end{array}\}.$$

At the far field when the distance of point P from the origin r is greater, such that $r\gg \frac{d^{2}}{\lambda }$, the difference in the amplitude of sound pressure is negligible. Therefore, the distance $r_{1}$, $r_{2},\;r_{3}$ from three array elements $S_{1}$, $S_{2}$, $S_{3}$ to point P respectively, can be replaced with the distance r. On the other hand, the phase difference caused by the different distances to the array elements cannot be ignored. 

Replacing the phase part of Equation (2) with Equation (3), we can obtain:(4)$$p=\frac{A}{\;r}e^{j\left(\omega t-kr\right)}\cdot \Psi ,$$
(5)$$\Psi =e^{j\left(kd\cos\left(\frac{0}{3}\cdot 2\pi -\varphi \right)\right)}+e^{j\left(kd\cos\left(\frac{1}{3}\cdot 2\pi -\varphi \right)+\frac{2}{3}\pi \right)}+e^{j\left(kd\cos\left(\frac{2}{3}\cdot 2\pi -\varphi \right)+\frac{4}{3}\pi \right)}.$$

Simplifying Ψ, and transforming the exponential term into a trigonometric function term according to Euler’s formula, we have:(6)$$\begin{array}{cc} \Psi & =-\sin\left(\frac{\pi }{6}-dk\sin\left(\frac{\pi }{6}-\varphi \right)\right)-\sin\left(dk\sin\left(\varphi +\frac{\pi }{6}\right)+\frac{\pi }{6}\right)+\cos\left(dk\cos\left(\varphi \right)\right)+\\ & i\cos\left(\frac{\pi }{6}-dk\sin\left(\frac{\pi }{6}-\varphi \right)\right)-i\cos\left(dk\sin\left(\varphi +\frac{\pi }{6}\right)+\frac{\pi }{6}\right)+i\sin\left(dk\cos\left(\varphi \right)\right). \end{array}$$

Further expanding and rearranging, we can obtain:(7)$$\begin{array}{c} \Psi =\cos\left(dk\cos\left(\varphi \right)\right)+i\sin\left(dk\cos\left(\theta \right)\right)-\cos\left(\frac{1}{2}dk\cos\left(-\varphi \right)\right)\cos\left(\frac{1}{2}\sqrt{3}dk\sin\left(-\varphi \right)\right)\\ \;+i\left(\sin\left(\frac{1}{2}dk\cos\left(-\varphi \right)\right)\cos\left(\frac{1}{2}\sqrt{3}dk\sin\left(-\varphi \right)\right)\right)\\ \;-i\sqrt{3}\left(\sin\left(\frac{1}{2}\sqrt{3}dk\sin\left(-\varphi \right)\right)\sin\left(\frac{1}{2}dk\cos\left(-\varphi \right)\right)\right)\\ \;+\sqrt{3}(\sin\left(\frac{1}{2}\sqrt{3}dk\sin\left(-\varphi \right)\right)\cos\left(\frac{1}{2}dk\cos\left(-\varphi \right)\right)). \end{array}$$

After extracting the spiral phase term, we can finally obtain:(8)$$\begin{array}{c} \Psi =\frac{3}{2}idke^{i\varphi }+\left(\cos\left(dk\cos\left(\varphi \right)\right)-1\right)+idk\cos\left(\varphi \right)\left(\frac{\sin\left(dk\cos\left(\varphi \right)\right)}{dk\cos\left(\varphi \right)}-1\right)\\ \;+\frac{1}{2}idk\cos\left(\varphi \right)\left(\cos\left(\frac{1}{2}\sqrt{3}dk\sin\left(-\varphi \right)\right)-1\right)\\ \;+\frac{3}{2}dk\sin\left(-\varphi \right)\left(\cos\left(\frac{1}{2}dk\cos\left(\varphi \right)\right)-1\right)\\ \;+\frac{1}{2}idk\cos\left(-\varphi \right)\left(\frac{\sin\left(\frac{1}{2}dk\cos\left(\varphi \right)\right)}{\frac{1}{2}dk\cos\left(\varphi \right)}-1\right)\cos\left(\frac{1}{2}\sqrt{3}dk\sin\left(-\varphi \right)\right)\\ \;+\frac{3}{2}dk\sin\left(-\varphi \right)\left(\frac{\sin\left(\frac{1}{2}\sqrt{3}dk\sin\left(-\varphi \right)\right)}{\frac{1}{2}\sqrt{3}dk\sin\left(-\varphi \right)}-1\right)\cos\left(\frac{1}{2}dk\cos\left(\varphi \right)\right)\\ \;+\left(1-\cos\left(\frac{1}{2}dk\cos\left(\varphi \right)\right)\cos\left(\frac{1}{2}\sqrt{3}dk\sin\left(-\varphi \right)\right)\right)\\ \;-i\sqrt{3}(\sin\left(\frac{1}{2}\sqrt{3}dk\sin\left(-\varphi \right)\right)\sin\left(\frac{1}{2}dk\cos\left(\varphi \right)\right)). \end{array}$$

Thus, the spiral wave field component $p_{1}$ of the total sound field can be written as:(9)$$p_{1}=\frac{3}{2}\frac{A}{r}e^{j\left(\omega t-kr\right)}idke^{i\varphi },$$
and the interference wave component $p_{2}$ can be written as:(10)$$p_{2}=p-p_{1}.$$

It can be noted that the spiral sound field component $p_{1}$ contains the phase component $e^{i\varphi }$, that changes linearly with azimuth. 

### 2.2. The Performance of the Spiral Sound Field

The sound field of phased array contains the spiral sound field component $p_{1}$ and the interference component $p_{2}$, along with a big phase characteristic difference between $p_{2}$ and $p_{1}$, leading to fluctuation in phase directivity. The relationship between the performance of the spiral sound field and the product of the array radius and wavenumber is discussed from the variation of the relative magnitudes of two components with dk (the product of array and radius wavenumber).

The proportion of the interference component in the total sound field in the far field can be written as:(11)$${\mathrm{Ratio}}_{1}\left(\varphi \right)=\frac{\left|p_{2}\right|}{\left|p\right|}.$$

Figure 2 shows a variation diagram of ${\mathrm{Ratio}}_{1}\left(\varphi \right)$ with dk. It can be seen that the proportion of the interference component in the total sound field increases with the increase in dk. When dk is very small, the proportion of the interference component in the total sound field is very small, whereas the proportion of the spiral sound field component in the total sound field is very large. Then, with the increase of dk, there is a gradual increase in the proportion of the interference component, and a gradual decrease in the proportion of the spiral sound field component in the total sound field.

The phase directivity function of the sound field of phased array can be written as:(12)Phase(φ)=arg(p).

The slope of the phase directivity function of the sound field of phased array can be written as: (13)$$\mathrm{KL}\left(\varphi \right)=\frac{d}{d\varphi }\mathrm{Phase}\left(\varphi \right).$$

Figure 3 shows a variation diagram of the phase directivity, and the slope of the phase directivity curve with dk. It can be seen from Figure 3 that when dk is very small, the phase directivity of phased array has a good linearity, and the slope of the phase directivity has a small amplitude fluctuation around 1. Then, with the increase in dk, the nonlinear fluctuation of the phase directivity of phased array increases gradually, and the fluctuation of the slope of the phase directivity increases gradually.

The ratio between the sound pressure amplitude of a phased array sound field and the sound pressure amplitude of the sound field produced by a single transducer can be written as: (14)$${\mathrm{Ratio}}_{2}\left(\varphi \right)=\frac{\left|p\right|}{\left|p_{E}\right|}.$$

Figure 4 shows a variation diagram of ${\mathrm{Ratio}}_{2}\left(\varphi \right)$ with dk, and it can be seen that, when dk is very small, the amplitude of the phased array sound field is very small, whereas the amplitude directivity of the sound field is not obvious. Then with the increase in dk, the amplitude of the phased array sound field increases, and the amplitude directivity of the sound field increases.

Therefore, we can conclude that in the range of dk∈(0,2), when dk increases, the amplitude of the phased array sound field increases, the nonlinear fluctuation of phase directivity of the sound field increases, the phase directivity increases with the fluctuation of the azimuth rate, and the amplitude directivity becomes higher directivity rather than omni-directivity.

### 2.3. Finite Element Simulation of Phased Array

The model of phased array in water was created in the finite element software COMSOL, to analyze the sound field produced by phased array. The simulation runs on a Windows PC. Figure 5 shows the finite element model of phases array.

The Pressure Acoustic, Frequency Domain Interface of COMSOL was used in the simulation. The monopole domain sources parameters were $Q_{m}$ are 1, $e^{\frac{2}{3}i\pi }$, $e^{\frac{4}{3}i\pi }$ respectively, for the three monopole sources. By adjusting the frequency parameter in the Study node, the sound field of the phased array with different dk was analyzed.

Figure 6 and Table 1 compare the simulation result of phase directivity and amplitude directivity with the theoretical calculation results.

It can be observed from Figure 6 that when dk = 0.73, the theoretical calculation of the amplitude range in each direction and the finite element simulation analysis value are 3.17 dB and 3.22 dB respectively, with an error of 0.05 dB, and the theoretical calculation and finite element simulation of the phase directivity fluctuation are 20.39° and 20.82° respectively, with an error of 0.1%.

This shows that the theoretically-derived results can directly be used to describe the amplitude directivity and phase directivity of the phased array in water. 

## 3. Spiral Sound Source Based on Phased Array

### 3.1. Basic Structure of Spiral Sound Source

From the theoretical and finite element simulation results, we can see that the amplitude of sound pressure can be increased with the increase in the product of array radius and wave number, with a cost of increase in phase directivity fluctuation, and increase of the amplitude directivity of the sound field. Given that the phase directivity fluctuation of the sound field is not very large, to keep the response of the sound source at a decent level, we designed the spiral wave sound source based on phased array with three transducers.

The spiral sound source consists of three transducers and a support. The transducer is a spherical piezoelectric ceramic transducer used for emitting omni-directional spherical waves. The array element transducers are fixed at three vertexes of an equilateral triangle by the support. Three signals are connected to three array element transducers respectively, for driving the sound source with the same frequency and same amplitude but phase difference of $\frac{2}{3}\pi ,$ respectively.

### 3.2. Finite Element Simulation of the Spiral Sound Source

The commercial finite element software COMSOL was used to establish the finite element model of the transducer [14,15,16]. COMSOL software was used to analyze the emission performance of the spiral sound source. Different from the simulation in 2.3, this simulation uses The Acoustic-Piezoelectric Interaction, Frequency Domain Interface, to simulate the spiral wave sound source. As the structure of the spiral sound source is symmetrical, a half model of the sound source was adopted to carry out the emitting simulation of a spiral sound wave in water. PZT-5J material was chosen for the piezoelectric ceramic spherical shell of the array element transducers with a diameter of 30 mm and thickness of 1mm. The influence of the support and other structures on the spiral sound source was ignored. Figure 7 shows the finite element model of the spiral sound source.

The paper studies the low-frequency emission characteristics for the spiral sound source, because dk is very large at resonance, causing series fluctuation of the phase directivity. The transmitting voltage response curve for the spiral sound source is shown in Figure 8. When the frequency is 3.5 kHz, the transmitting voltage response is 108.9 dB, and the transmitting voltage response of a single transducer is 106.8 dB.

The simulation curve of the rate of change of the phase directivity of the sound source with the azimuth is shown in Figure 9. It can be seen that the rate of change of the phase directivity with the azimuth varies from 0.53 to 1.70, and is always greater than 0, indicating a continuous monotone variation of the phase directivity with the azimuth.

The finite element simulation results of the phase directivity for the sound source is shown in Figure 10. It can be seen that as the azimuth is increased by 360°, the phase of the sound field is also increased by 360°, indicating that the sound field of the spiral sound source is a spiral sound field, and the nonlinear fluctuation of the phase directivity is ±16°.

The finite element simulation result of the amplitude directivity for the sound source is shown in Figure 11. It can be seen that the amplitude directivity of the sound source is periodic with a range of 3.4 dB.

According to the finite element simulation analysis, we conclude that the spiral wave sound source based on phased array with three transducers can emit the spiral sound wave where the phase continuously varies with the azimuth, such that when the azimuth varies by 360°, the phase of the sound field also varies by 360°.

## 4. Measurement of the Spiral Wave Source

### 4.1. Measuring Principle of Phase Directivity

If the sound source is placed at the origin of coordinates, the two hydrophones are placed at $x_{1}\left(r_{0},0,\varphi_{1}\right)$ and $x_{2}\left(r_{0},0,\varphi_{2}\right)$ respectively, then the azimuth difference can be written as:(15)$$\varphi_{2}-\varphi_{1}=\Delta \varphi_{1}.$$

If the two hydrophones collect sound pressure signals at the same time, the signals collected are given as:(16)$$s_{1}=M_{e}Pe^{j\left(\omega t-kr_{0}\right)}e^{j\mu \varphi_{1}},$$
(17)$$s_{2}=M_{e}Pe^{j\left(\omega t-kr_{0}\right)}e^{j\mu \varphi_{2}}.$$

The phases of two signals at t=0 are then:(18)$$Arg_{1}=-kr_{0}+\mu \varphi_{1}$$
(19)$$Arg_{2}=-kr_{0}+\mu \varphi_{2}$$

The phase difference of the output signal between the two hydrophones can be written as:(20)$$Arg_{2}-Arg_{1}=\Delta \varphi_{2}$$

The sound field can be regarded as a spiral wave field if the experimental results prove that the phase difference $\Delta \varphi_{2}$ between the output signals of two hydrophones equals the azimuth difference $\Delta \varphi_{1}$ of two hydrophones. 

To calculate the curve of the ratio KL with the azimuth, we first rotated the sound source, and then measured the phase difference $\Delta \varphi_{2}$ between the output signals of two hydrophones in different directions. By integrating KL to φ, the phase directivity curve of the measured sound field was obtained.

### 4.2. Test Results of the Spiral Sound Source

A prototype of the spiral wave sound source based on phased array with three transducers was designed and fabricated according to the finite element simulation results. Figure 12 is the picture of spherical transducers and spiral sound source prototype. The resonant frequency of the array element transducer of the spiral sound source is 43 kHz, and the diameter of the transducer is 37.8 mm. Three transducers are fixed by the support which enables the three transducers to be both isometrically and circumferentially distributed, and the distance between the center of the support chuck and the center of the support is 50 mm.

Figure 13 shows the placement of the spiral wave sound source and hydrophones in the pool and equipment connection. The fabricated spiral wave sound source was tested in a reverberation pool. By measuring the direct pulse that the hydrophones received, the free field performance of the spiral wave sound source was tested. Two signal sources outputs three pulsed signals, where the phase difference of each signal is realized by triggering time delay. All the three signals were amplified to 200 Vpp through 3 power amplifiers and connected to the transducers. The frequency of the signal is f=3.5 kHz, and time delays are $0,\frac{1}{3f},\;\frac{2}{3f}$, respectively. The models of equipment involved are shown in Table 2.

The experimental result of the transmitting voltage response of array element transducer is shown in Figure 14. The transmitting voltage responses of three array element transducers at a frequency of 3.5 kHz are 106.5 dB, 105.9 dB, 105.3 dB respectively, with a range of 1.2 dB, indicating good uniformity and conforming the simulation result.

The experimental results of the horizontal phase directivity slope KL-curve of the spiral sound source are shown in Figure 15. KL fluctuates between 0.35 and 2.11 and is constantly above 0, indicating a monotone increase of the sound field phase with the increase of the phase. Compared with the simulation, we can see that the change trend of the slope of the phase directivity with the azimuth is basically in agreement with the simulation results.

The test results of the horizontal phase directivity curve of the spiral sound source are shown in Figure 16. It can be seen that when the azimuth is increased by 360°, the phase of the sound field is also increased by 360°, indicating that the sound field of the spiral sound source is a spiral sound field, and the nonlinear fluctuation measurement value of phase directivity is ±21°.

The test results of the horizontal amplitude directivity diagram of the spiral sound source are shown in Figure 17, it can be seen that the range of amplitude directivity in all directions is 4.3 dB.

The test results show that the spiral wave sound source based on phased array with three transducers can emit spiral sound waves with the phase continuously varying with the azimuth, and the measured results are in agreement with the simulation results.

## 5. Conclusions

In this paper, the phased array with three omni-directional transducers was researched to apply in the emission of underwater spiral sound waves as a simple realization form of underwater spiral sound source. According to the superposition theory of sound field, the mathematical derivation of far-field sound pressure distribution of the sound field produced by phased array was given, the generation of spiral sound wave emitted by the phased array was obtained, and the influence of array parameters on phase directivity, amplitude directivity, and amplitude of the sound field were discussed. On this basis, the spiral wave sound source based on phased array which takes the spherical piezoelectric ceramic transducers as the array elements was analyzed with finite element software simulation. The sound source was made, and the sound field was measured. 

The theoretical, simulation and experimental results show that the phased array with three transducers can emit spiral waves, the performance of the phased array was affected by the product of array radius and wave number. When the product of array radius and wave-number increases, the sound field amplitude reduces, and the phase directional fluctuation decreases. The phase of the sound field produced by the prototype of the spiral sound source increases continuously with the azimuth at 3.5 kHz. The phase directional fluctuation was ±21°. The measurement results of amplitude directivity and phase directivity are in agreement with the theoretical and simulation results, verifying the correctness of the generation of spiral wave by the phased array, and the realization of the spiral wave sound source.

## Figures and Tables

**Figure 1 sensors-19-03192-f001:**
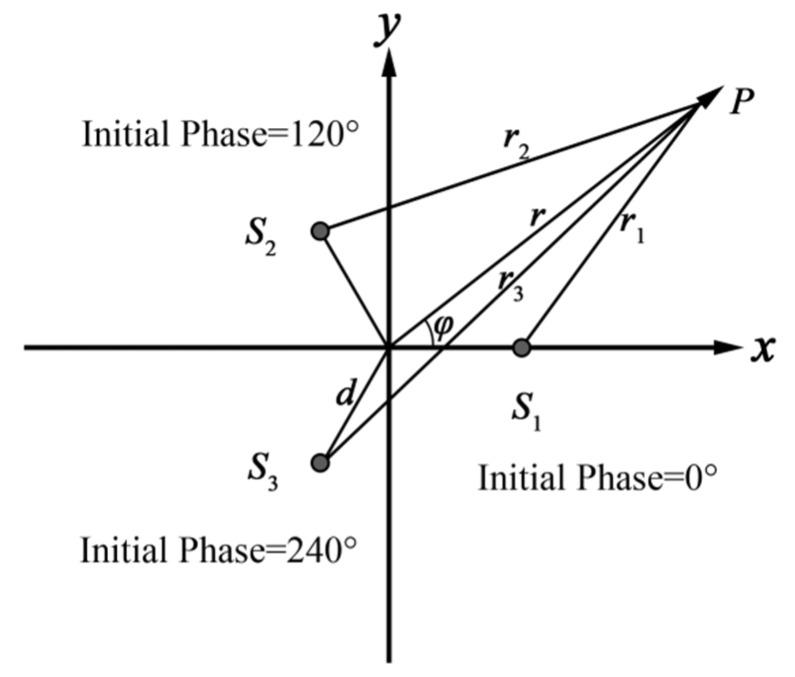
Phased array with three transducers and coordinate system.

**Figure 2 sensors-19-03192-f002:**
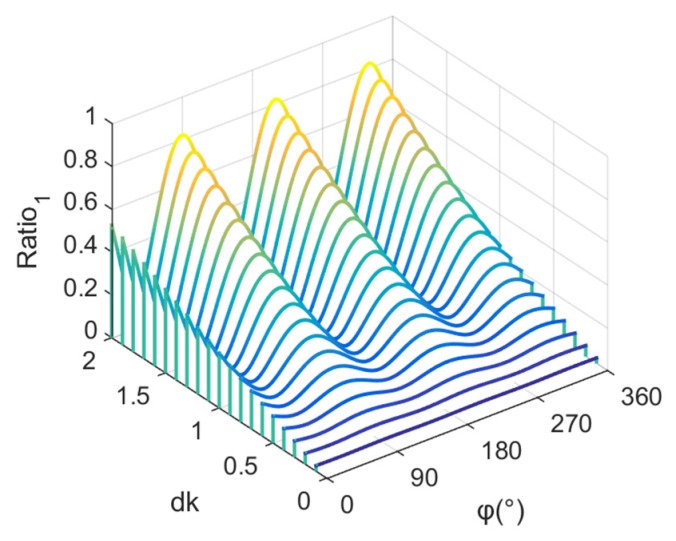
Variation of the proportion of the interference component in the total sound field with dk.

**Figure 3 sensors-19-03192-f003:**
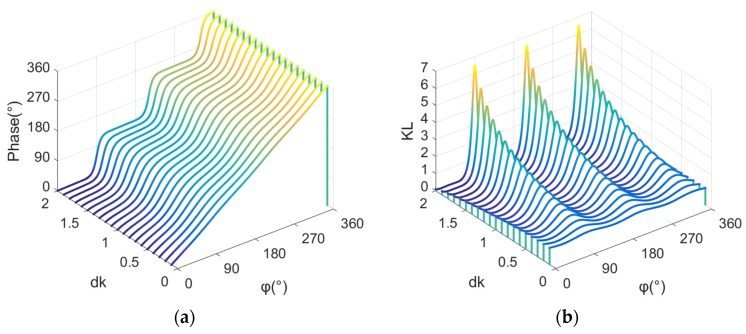
(**a**) Variation of the phase directivity of phased array with dk. (**b**) Variation of the slope of the phase directivity of phased array with dk.

**Figure 4 sensors-19-03192-f004:**
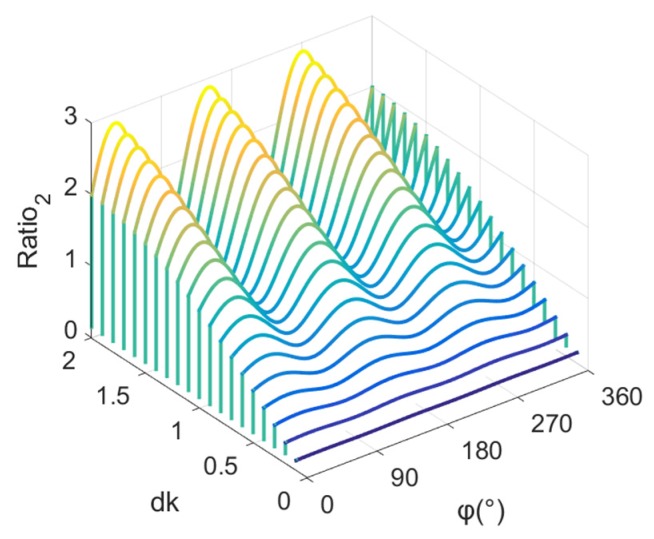
Variation of the ration between the sound pressure amplitude of a phased array sound field and the sound pressure amplitude of the sound field produced by a single transducer with dk.

**Figure 5 sensors-19-03192-f005:**
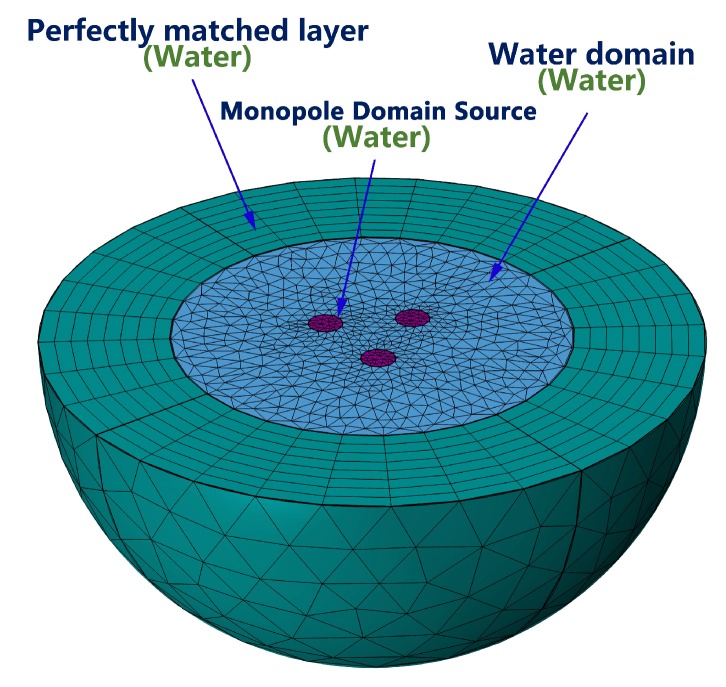
Finite Element Simulation Model of Phased Array.

**Figure 6 sensors-19-03192-f006:**
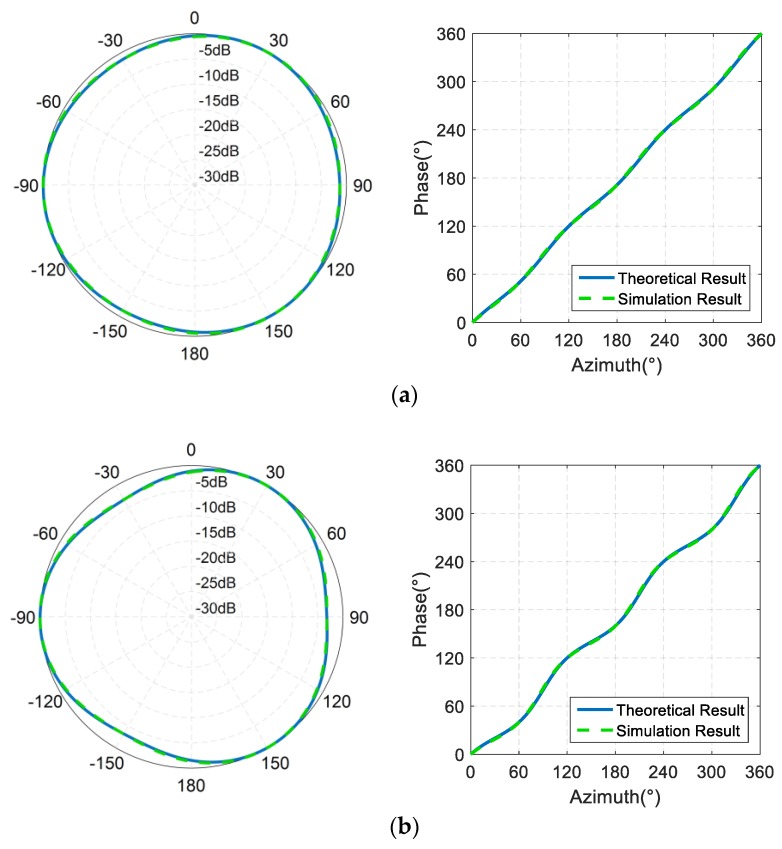
Comparison of the finite element simulation with the theoretical calculation results. (**a**) Amplitude directivity and phase directivity of phased array sound field, where dk=0.3. (**b**) Amplitude directivity and phase directivity of phased array sound field, where dk=0.73. (**c**) Amplitude directivity and phase directivity of phased array sound field, where dk=1.6.

**Figure 7 sensors-19-03192-f007:**
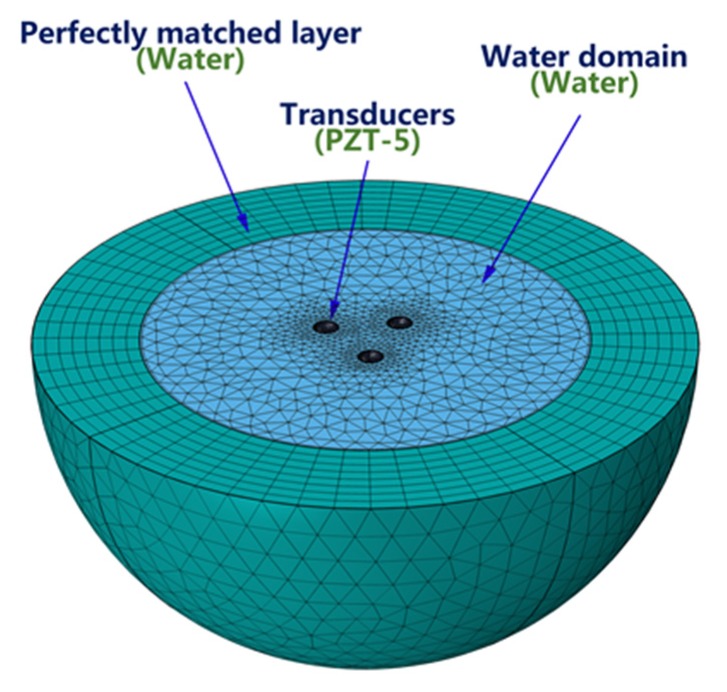
Finite element model for the spiral sound source in water.

**Figure 8 sensors-19-03192-f008:**
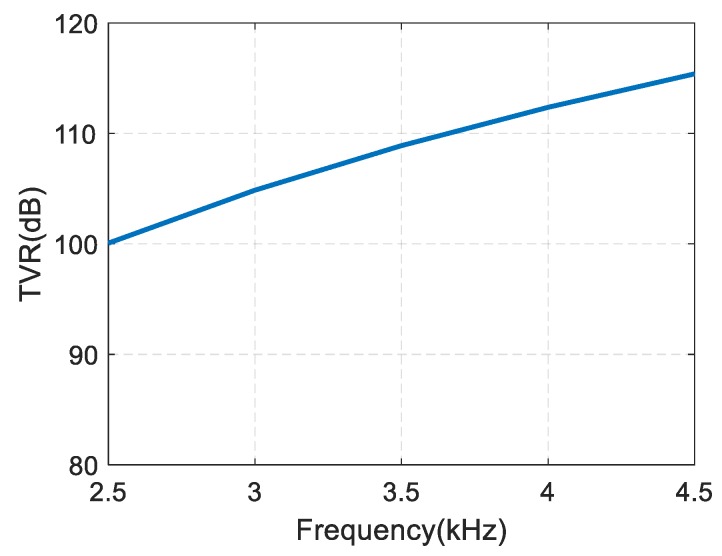
Finite element simulation results of transmitting voltage response for the spiral sound source.

**Figure 9 sensors-19-03192-f009:**
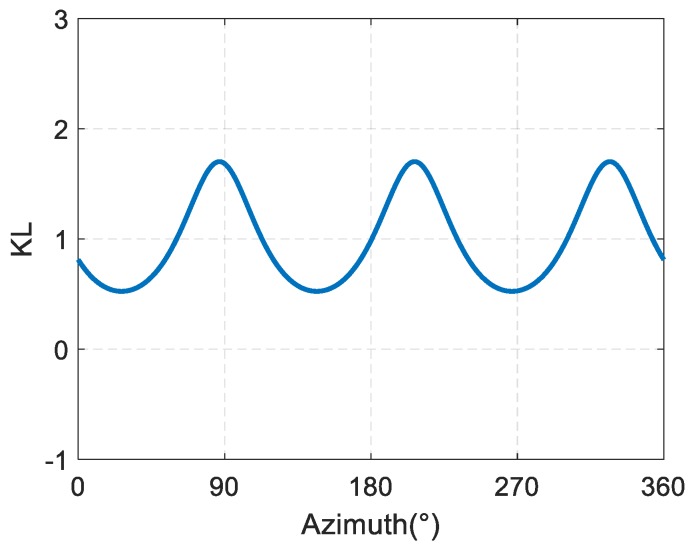
Finite element simulation results of the change rate of the phase directivity for the spiral sound source with the azimuth.

**Figure 10 sensors-19-03192-f010:**
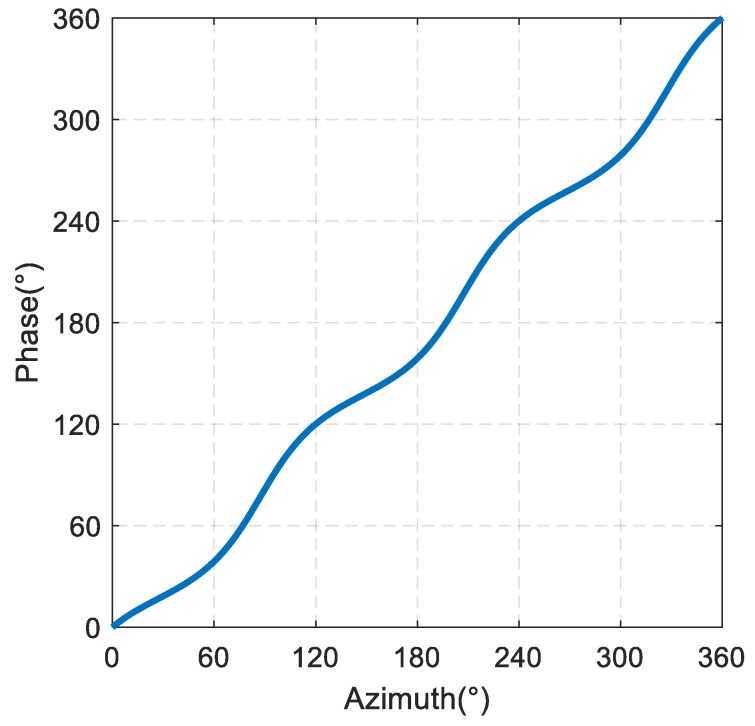
Finite element simulation results of the phase directivity for the spiral sound source.

**Figure 11 sensors-19-03192-f011:**
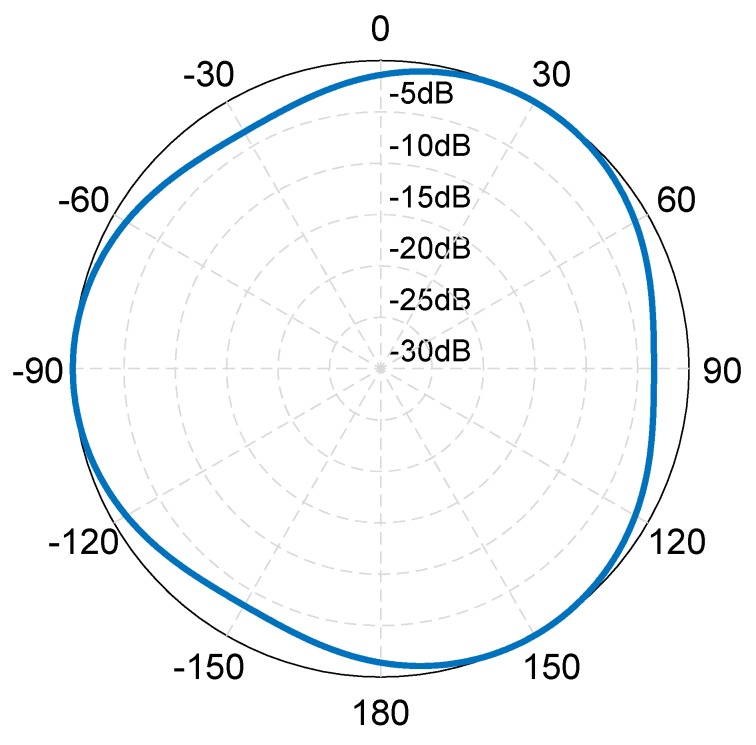
Finite element simulation results of the amplitude directivity for the spiral sound source.

**Figure 12 sensors-19-03192-f012:**
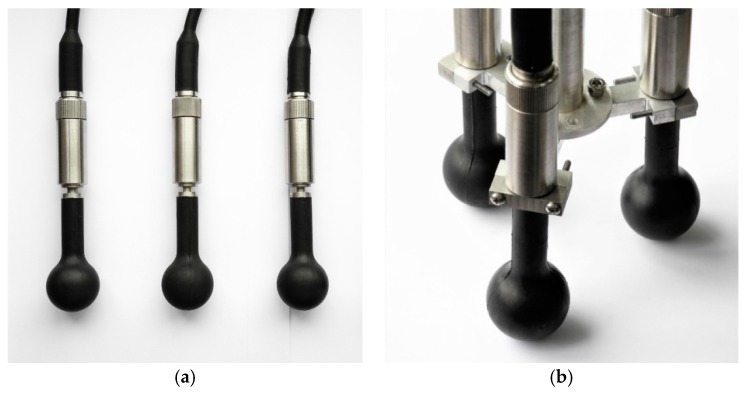
(**a**) Spherical transducers and (**b**) Spiral sound source after assembly.

**Figure 13 sensors-19-03192-f013:**
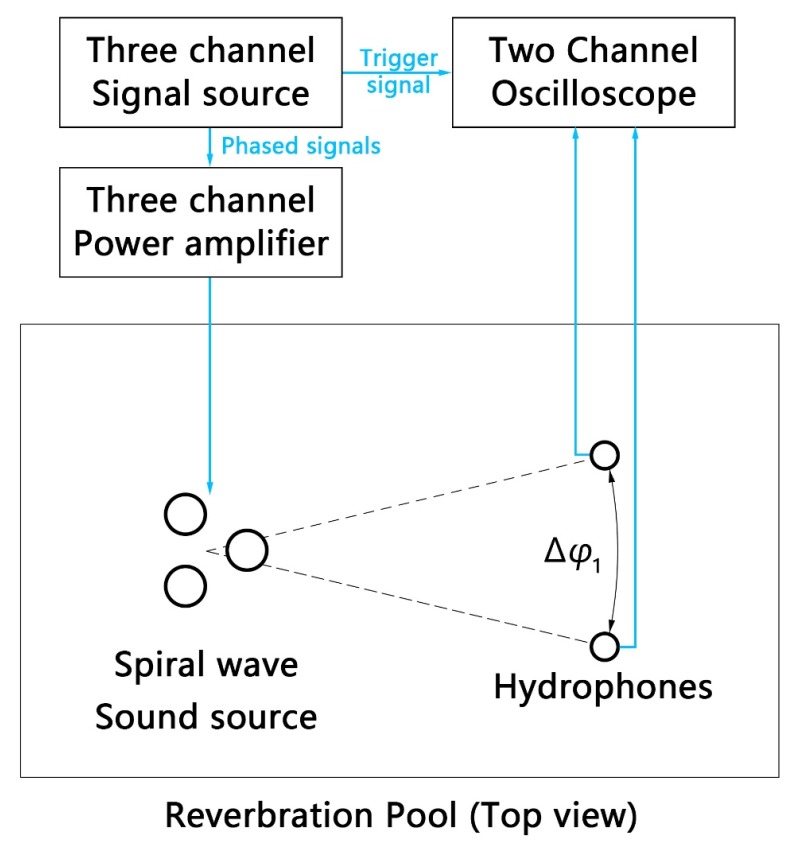
Sketch diagram of testing equipment and equipment connection.

**Figure 14 sensors-19-03192-f014:**
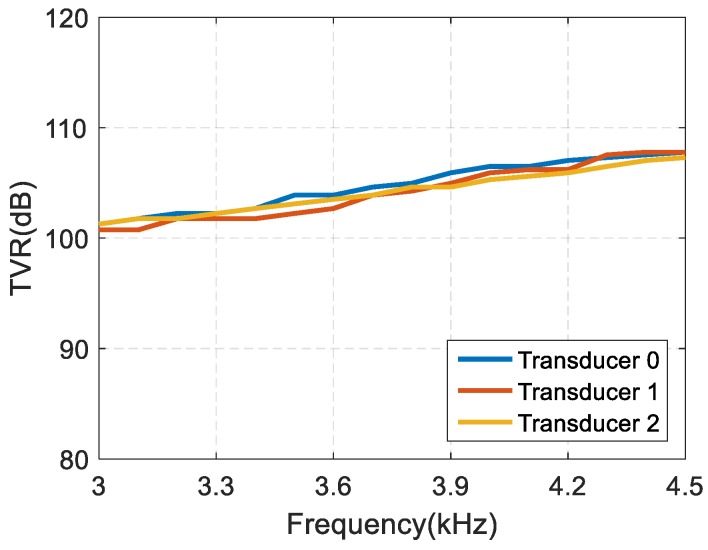
Test results of the transmitting voltage response of array element transducer.

**Figure 15 sensors-19-03192-f015:**
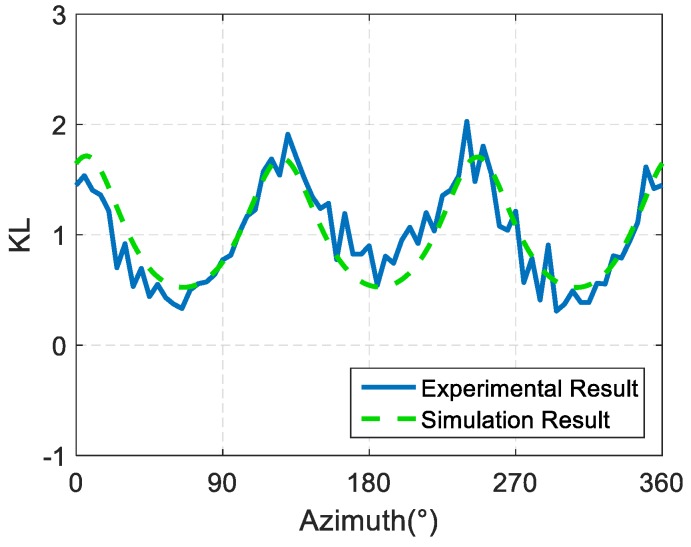
Test results of horizontal phase directivity slope KL-curve of the spiral sound source.

**Figure 16 sensors-19-03192-f016:**
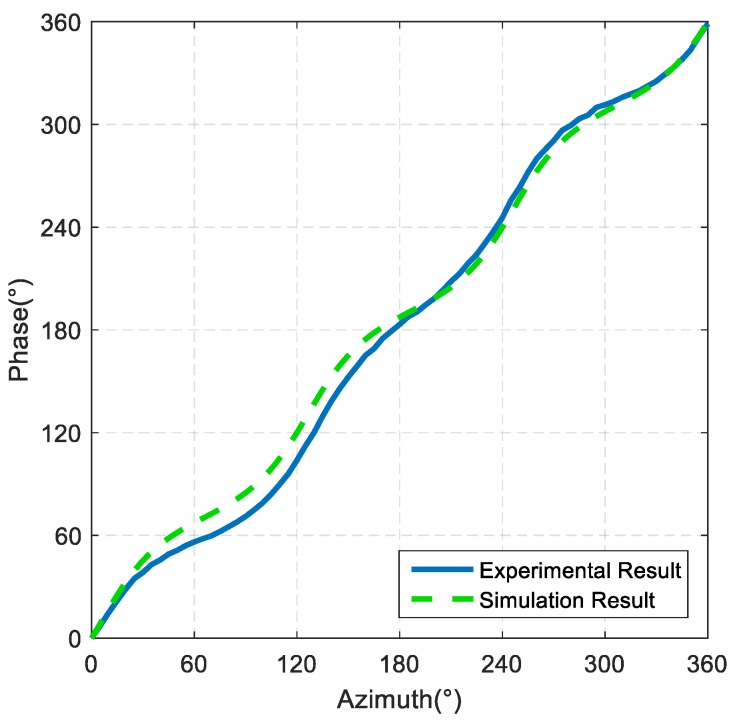
Test results of the horizontal phase directivity curve of the spiral sound source.

**Figure 17 sensors-19-03192-f017:**
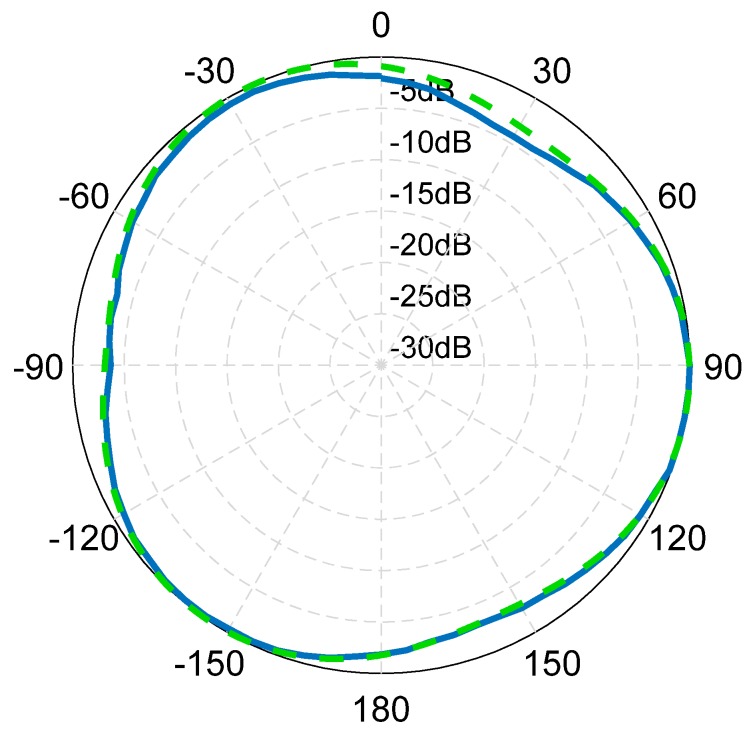
Test results of horizontal amplitude directivity diagram of the spiral sound source.

**Table 1 sensors-19-03192-t001:** Comparison of the theoretical calculation with the finite element calculation results (dk = 0.73).

	Theoretical Calculation	Finite Element Simulation
Amplitude Directivity Range	3.17 dB	3.22 dB
Phase directivity Fluctuation Range	20.39°	20.82°

**Table 2 sensors-19-03192-t002:** Model of equipment used in measurement.

Name	Type	Number (pcs)
Signal Source	RIGOL DG4062	2
Power Amplifier (1)	INSTRUMENTS Model L6	2
Power Amplifier (2)	INSTRUMENTS Model L2	1
Hydrophone	B&K 8105	2
Oscilloscope	Tektronix DPO 2014	1

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
