# Peer review of "Underwater Spiral Wave Sound Source Based on Phased Array with Three Transducers"

_sensors, 2019, doi:10.3390/s19143192_

Reviewer 1 Report

In my opinion, the work deals with interesting issues, is prepared quite carefully and presents the results of experimental research supported by computer-assisted analyzes and analytical calculations. The authors showed a good agreement on the results obtained. However, the article requires amendments that will improve its quality. The work may be interesting for other researchers dealing with similar issues.

Remarks:

1. A sentence/name: „A spiral wave sound source based on phased array with three spherical piezoelectric ceramic transducers” is repeated in the text too many times (espetially in the abstract). This is not good for the readability of the article in my opinion. A good idea would be to introduce some shortcut.

2.The authors should extend the introduction to the work and discuss the achievements of other researchers in this field, pointing out the essence of their own achievements in this context. This section should be significantly expanded.

3.The chapter „2.3. Finite Element Simulation of Phased Array” should be extended. Which computer system was used for the finite element analysis? Please extend and provide more details.

4.A minor text correction is required - for example, dots should be placed after references to literature, in some places there are lacks of spaces.

5.Punctuation should be improved before (all equations) and after (equations 3,13,14) equations- they are parts of sentences.

6.In the reference numer 16 names were used instead of authors' surnames(Konrad, D.; Krzysztof, S.; Andrzej, B.) - it must be corrected.

Reviewer 2 Report

This manuscript is totally very well written. However, following remark should be taken into account to be published.

 Remark:

2.3. Finite Element Simulation of Phased Array

Page 6, line 2

The model of phased array in water is created in the finite element software...

The model is not clear. Authors should explain more in detail. For example, the boundary condition and analytical area like as Figure 6 of “3.2. Finite Element Simulation of the Spiral Sound Source”.

 Author Response

Reviewer 3 Report

In this paper, a spiral wave sound source based on phased array with three spherical piezoelectric ceramic transducers is then designed and fabricated based on FEM simulation, and the performance of the sound source is analyzed. Firstly, the far-field   sound pressure distribution of the sound field produced by phased array is analyzed. On this basis, the spherical piezoelectric ceramic transducers as the array elements are analyzed with finite element software simulation.

 There are still some problems in the paper, which need to be pointed out:

1. There are two identical numbers in Figure 7 of the paper. The numbering of Figure 8 is missing and the indication is not clear in the text.

2. The innovation of the paper is slightly insufficient, Only a spiral wave sound source is designed, and the innovation of theoretical methods is lacking.

3. The hardware design of the spiral wave sound source is not introduced enough in this paper. The performance comparison of the same type of products is lacking only in theoretical verification and analysis.

4. The experimental analysis of the lack of actual scenes, the practical performance of the designed product is not given.

Author Response

Round  2

Reviewer 3 Report

The authors have corrected all questions.